# Sleep deprivation disrupts diurnal rhythmicity of gut microbiota and blood inflammatory cytokines in mice

**Weiran Shan, Wenzhe Zang, Zhiyi Zuo** *

Department of Anesthesiology, University of Virginia, Charlottesville, Virginia, United States of America

* zz3c@virginia.edu

## Abstract

### Background

Disruption of diurnal rhythms can lead to various diseases in humans and animals. There is an interaction between the diurnal rhythms of host and gut microbiota. This study was designed to determine whether sleep deprivation disrupted the diurnal rhythmicity of microbiota in the ileum, blood inflammatory cytokines and blood short chain fatty acids (SCFAs). SCFAs are products of gut microbiota.

### Methods

Six- to eight-week old CD1 male mice were sleep-deprived for 24 h by placing them in a small platform in a water tank. Their blood and ileal samples were harvested every 4 h in the next day. Mice with or without sleep deprivation were subjected to open field test.

### Results

Mice with sleep deprivation had decreased richness and altered taxonomic composition of microbiota in the ileum compared with controls. Sleep deprivation disrupted the diurnal rhythmicity of gut microbiota and blood inflammatory cytokines. Mice with sleep deprivation had higher cytokine concentrations (for example, interleukin 1β concentrations at Zeitgeber time 21 were 8.4±0.9 and 5.4±0.4 pg/ml for sleep deprivation and control groups, respectively, P=0.0128) and were more anxious as assessed by open field test than control mice. Sleep deprivation did not affect the concentrations of SCFAs in the blood.

### Conclusions

Our results suggest that there are diurnal rhythms in the microbiota of ileum and blood inflammatory cytokines. Sleep deprivation disrupts these rhythms. Considering

**Data availability statement:** The 16S rRNA sequencing data of this study can be accessed via https://www.ebi.ac.uk/biostudies/arrayexpress/studies/E-MTAB-15462?key=646b5644-31ad-464e-9d83-6cd081340bcf at the EMBL-EBI site. All other data are available via https://doi.org/10.6084/m9.figshare.29791706.v1 at the Figshare site.

**Funding:** This study is funded by the Robert M. Epstein Professorship endowment, which is a professorship I have held. This professorship belongs to the University of Virginia, my employer. Obviously, my university had no role in study design, data collection and analysis, decision to publish, or preparation of the manuscript. There was no external funding to support our study.

**Competing interests:** The authors have declared that no competing interests exist.

the known broad actions of inflammation, the increased inflammatory cytokines may mediate various biological effects, such as anxiety, of sleep deprivation.

## Introduction

Circadian rhythms are common phenomena in biological systems. One cycle of circadian rhythm is about 24 h [1]. The circadian rhythm is controlled by cells in specific brain regions, such as suprachiasmatic nucleus, and regulated by signals from various peripheral systems, such as digestive system, to modulate the expression of circadian clock genes in the brain [2–4]. Maintaining normal circadian rhythms is important for physiological functions including immune functions. Disturbance of circadian rhythms can lead to obesity, diabetes, cancer and cardiovascular diseases [5].

Interestingly, abundant evidence in the last decade has shown that gut microbiota has diurnal rhythms [2,5]. There is a significant interaction between the circadian rhythms of the host and the diurnal rhythms of gut microbiota. For example, feeding of the host, a behavior that is a presentation of circadian rhythms, is a strong factor to modulate the diurnal rhythms of gut microbiota [6,7]. The products of gut microbiota, such as short chain fatty acids (SCFAs), can regulate the expression of circadian clock genes [2]. Abnormal diurnal rhythms of gut microbiota can disturb basic biological activities, such as sleep [2].

Sleep is a basic component of diurnal rhythms of mammalian animals. Sleep disturbance is common in humans. About 200 million people fly in and out of the USA via international flights every year [8] and 15% of full-time workers in the USA perform shift work [9]. These people have sleep disruption. This disruption leads to abnormal gut microbiota. Interestingly, the transplantation of feces from people with jet lag to mice results in an increase of body weights in mice [10], providing evidence for the involvement of gut microbiota in mediating the effects of jet lag on metabolism.

It is clear that sleep deprivation can affect the gut microbiota in humans and animals [2]. However, it is unknown whether sleep deprivation affects the diurnal rhythms of gut microbiota. In addition, previous studies on gut microbiota often analyze fecal microbiota [10–14]. This sampling is prone to contamination and the microbiota in the feces may not reflect the microbial composition on the intestinal mucosa, which may be more important for the interaction with the host systems than the fecal microbiota. Thus, we hypothesize that there are diurnal rhythms of microbiota in the ileum and that these diurnal rhythms are disrupted by sleep deprivation in mice. To determine these hypotheses, gut microbiota in the ileum was analyzed every 4 h for a total of 24 h. In addition, the diurnal rhythms of SCFAs and representative proinflammatory cytokines in the blood were determined to understand the potential effects of gut microbiota rhythmicity on the host.

## Methods and materials

The animal protocol was approved by the Institutional Animal Care and Use Committee of the University of Virginia (Charlottesville, VA). All animal experiments were carried out in accordance with the National Institutes of Health Guide for the care and

use of laboratory animals (NIH publications number 80−23) revised in 2011. Efforts were applied to avoid any unnecessary sufferings of animals in the experiments. This manuscript was written to adhere to the ARRIVE guideline.

## Sleep deprivation

CD1 6- to 8-week-old male mice (purchased from Charles River International, Wilmington, MA, USA) that were kept on a 12/12 h light–dark cycle with ad libitum access to water and food were randomly assigned to two groups by computer-generated randomization table. The first group was control group that had no alterations in their living conditions. The second group was subjected to platform sleep deprivation. The platform was 5 cm in diameter and placed in a tank (40 × 30 cm) filled with water to within 1 cm of the upper surface of the platforms. Five mice from the same cage in which they were previously housed were placed in each tank for 24 h with water and food ad libitum [15]. The sleep deprivation started at Zeitgeber time (ZT) 1 (corresponding to 8:00 am of Eastern Standard Time) and ended the next day at ZT1. Blood and ileal feces were harvested the next day at ZT1, ZT5, ZT9, ZT13, ZT17 and ZT21 from both groups (n = 6 mice for each group at each time point) after being euthanized by isoflurane overdose. The sample size for each group was determined based on our experience and similar types of experiments and analyses in previous studies.

## ELISA of interleukins and corticosterone in plasma

Plasma was prepared from the blood and was used for enzyme-linked immunosorbent assay (ELISA) to assess the amount of interleukin (IL)-6, IL-1β, IL-17 and corticosterone as we described previously [16,17] by using the following kits: mouse IL-6 Quantikine ELISA kits (catalogue number: SM6000B, R&D SYSTEM), mouse IL-1β/IL-1F2 Quantikine ELISA kits (catalogue number: SMLB00C, R&D SYSTEM), mouse IL-17 Quantikine ELISA kit (catalogue number: M1700, R&D SYSTEM) and corticosterone parameter assay kit (catalogue number: KGE009, R&D SYSTEM).

## Assay of plasma SCFAs

SCFAs in the plasma were determined by a method similar to that reported before [13,18]. To every 200 µl plasma, 2 ml 0.05 M HCL containing 0.15 M NaCl was added. The solution was mixed with 8 ml extraction buffer (hexane: diethyl ether = 1:1) on a shaker for 20 min. After being centrifuged at 1800 g for 5 min, 7.5 ml supernatant was collected, mixed with 93 µl 20 mM KOH in methanol and dried at 40°C under nitrogen gas. The residue was reconstituted in 40 µl 2.5% 18-Crown-6 dissolved in acetonitrile and was further derivatized at 75°C for 30 min with the addition of 9-chloromethylanthracene and tetramethylammonium hydroxide in acetonitrile. Finally, 30 µl derivatization solution was loaded to Acclaim C18 column (3 µm, 4.6 × 100 mm, Thermo Scientific) and separated by Ultimate 3000 high performance liquid chromatograph (Thermo Scientific) equipped with an UV-visible detector. The peaks of derivatized SCFAs were detected at a wavelength of 254 nm. 2-Ethylbutyric acid (2-EA) was added as an internal reference control. The peak areas of acetic acid, propionic acid, butyric acid and valeric acid were measured as mAU*min and the peak area of each sample was normalized with that of 2-EA in the sample. Results were presented as ratios of acetic acid, propionic acid, butyric acid and valeric acid to 2-EA.

## 16s rDNA sequencing and gut microbiota profiling

Microbiota composition from ileal contents was tested by 16s rDNA sequencing as we described before [14,19]. Briefly, ileal contents were collected and stored at −80°C until further processing. Bacteria DNA was extracted with Power Lyzer Power soil DNA isolation kit (Catalogue number: 12855, QIAGEN, Germantown, MD, USA). Sequencing libraries of the hypervariable V3–V4 region were prepared according to the instructions of Illumina MiSeq system. Primers used for preparing the libraries were as follows: F: 5′-TCGTCGGCAGCGTCAGATGTGTATAAGAGACAGCCTACGGGNGGCWGCAG-3′ and R: 5-GTCTCGTGGGCTCGGAGATGTGTATAAGAGACAGGACTACHVGGGTATCTAATCC-3′. The amplicons were cleaned up with AMPure XP magnetic beads and then used for Index PCR by using the Nextera XT Index Kit. Qubit dsDNA HS

assay kit and TapeStation high sensitivity D1000 ScreenTape (Agilent, Blacksburg, VA, USA) were used to measure the concentrations of PCR products and normalize the quantity for library preparation. Sequencing was operated on an Illumina MiSeq instrument by MiSeq reagent kit v2 (500 cycles). Data was analyzed by Realgene (Shanghai, China) with the MiSeq Reporter software Metagenomics work flow v2.5.1.3 (Illumina, San Diego, CA, USA) as we did before [13].

The paired reads from double-terminal sequencing were spliced into a sequence by Pandaseq software. The long reads having high variable region were derived. The reads with an average phred score less than 20 in the window (5 bp in size, 1 bp step length) were trimmed. Reads having ambiguous "N" or with length shorter than 220 bp were not used. The high-quality sequences without chimeras were placed in order from large to small abundance and then clustered with 97% similarity into an Operational Taxonomic Unit (OTU). Each OTU was viewed as a species. To reduce the deviation of analysis resulted from the different sizes of sample sequencing data, the number of Reads to OTU was counted based on the minimum sequence number matched to OTU if the sequencing depth was sufficient. Random leveling was used to analyze alpha diversity. One Read from each OTU was used as a representative sequence. The representative sequence was used to identify bacterial species by comparing the sequence with ribosomal database project database. Species were classified for all OTUs, and species abundance tables were derived for subsequent analysis. OTU selection was done using Uclust on the software platform QIIME v1.9.1. Chao1 index, observed_species diversity index, Simpson index and Shannon index were used for comparing α diversity between groups. Principal coordinate analysis (PCoA) and analysis of similarity (ANOSIM) were used for β diversity analysis to compare microbial compositions among samples. The relative abundance of taxa within gut microbiota at genus level was compared between groups.

### Diurnal rhythm analysis

An in-house Matlab program based on cosinor analysis [20], which uses the least squares method to fit a sine wave to a time series, was developed for biorhythm analysis and prediction of interleukins, corticosterone and SCFAs in the plasma as well as microbiota in the ileum of both control and sleep deprivation groups. Key parameters including MESOR (*M*, *i.e.*, midline estimating statistic rhythm-adjusted mean), amplitude (*A*, *i.e.*, a measure of half the extent of predictable variation within a cycle), acrophase (*ϕ*, *i.e.*, a measure of the time of overall high values recuring in each cycle), and period (*τ*) were generated for each cosinor analysis. Zero-amplitude test (F-test) [20] was adopted to assess how well the circadian rhythm pattern fitted the data. P-value was calculated to determine the statistical significance with a pre-defined threshold as 0.05. The F-test is implemented using the MATLAB function "fpdf".

### Open field test

To determine the effects of sleep deprivation on mouse general behavior and to have a test with a low stress level, open field test was selected to be used. After sleep deprivation for 24 h, mice were allowed to rest for 5 h before open field test (n = 11–12 mice for each group). As we described before [21], control and sleep-deprived mice were placed in the open field box for 10 min. The time of mice spent in the corner, border and center areas were recorded and analyzed by ANY-maze behavioral tracking software (SD Instruments, Stoelting Co., IL). The performance and assessment of mice in the open field test was done in a blind fashion.

### Statistical analysis

We did not define specific inclusion and exclusion criteria before the experiments. Results from all mice that completed the measurements were included in the analysis. Parametric results in normal distribution were presented as means ± S.E.M. The data were analyzed by two-way analysis of variance with time and sleep deprivation as the two factors. Differences were considered significant at P < 0.05 based on two-tailed hypothesis testing. Statistical analyses were performed using the IBM SPSS statistics 27. Cosinor analysis was performed as described in section "Diurnal rhythm analysis". Gut microbiota was analyzed as described in section "16s rDNA sequencing and gut microbiota profiling".

## Results

### Sleep deprivation altered the microbiota and disrupted the diurnal rhythms of the microbiota in the ileum

First, we wanted to determine whether sleep deprivation affected the overall microbiota in the ileum. For this determination, microbiota results at different time points were pooled together for the analysis. Mice with sleep deprivation had a decrease in the α diversity in their microbiota compared with control mice (Fig 1A). These results suggest a decrease in the richness of microbiota species within each sample. The β diversity of control mice and mice with sleep deprivation was different (Fig 1B), suggesting that the species makeup of bacteria in the ileum of mice with sleep deprivation is different from that of control mice. The bacteria that were different in relative abundance between control and sleep deprivation groups are shown in Fig 1C. For example, the abundance of *g_alistipes* and *g_escherichia/shigella* in mice with sleep deprivation was higher than that in control mice and *g_anaerotruncus* was more abundant in the control mice (Fig 1C). These results suggest that sleep deprivation significantly changes the microbiota in the ileum.

We then wanted to determine the effects of sleep deprivation on the diurnal rhythmicity of microbiota. We decided to perform the analyses at order level to reduce the number of analyses comparing with the situation that analyses were

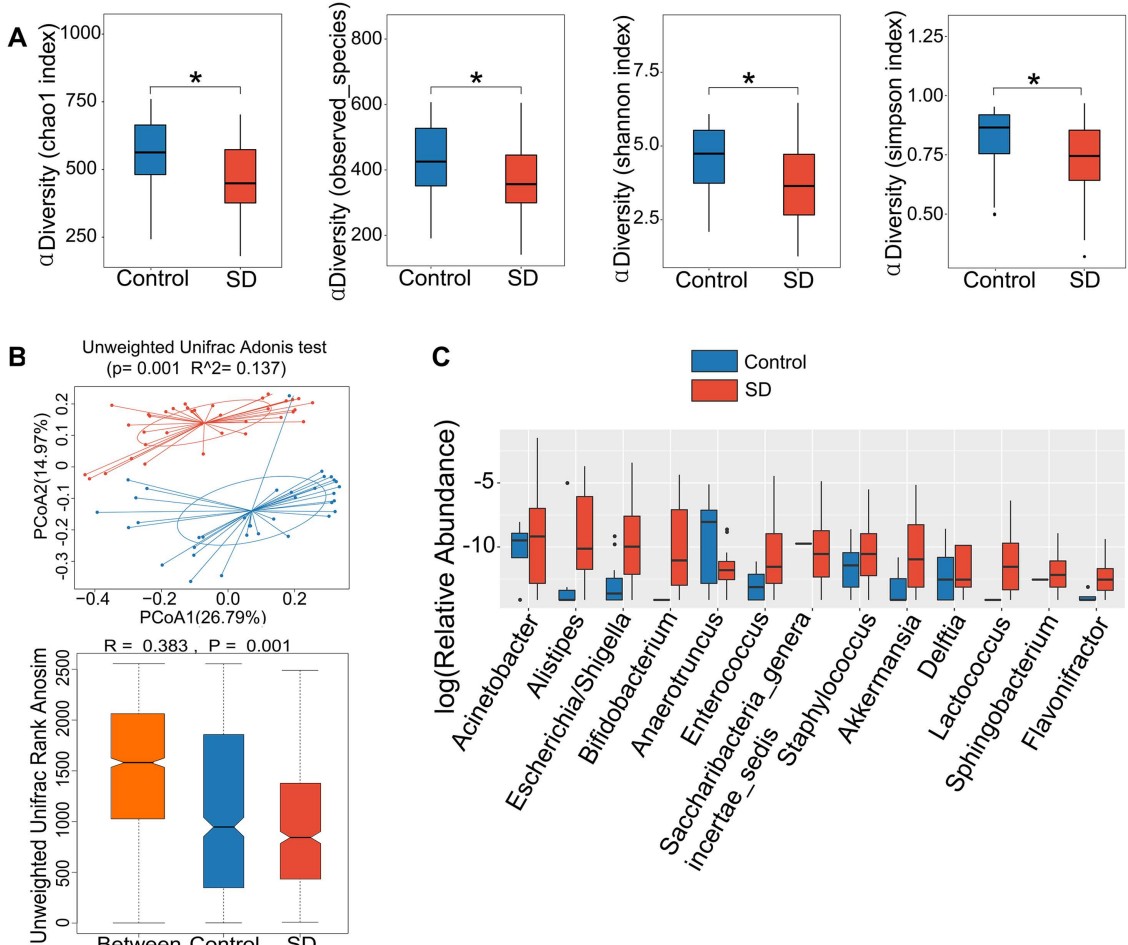

**Fig 1. Sleep deprivation altered the microbiota in the ileum.** (A) comparison of α diversity between groups. * P<0.05 for the comparisons. (B) comparison of β diversity between groups. (C) relative abundance of various bacteria at genus level. Only bacteria that were different in relative abundance between control and SD groups are shown here. The sample size was 6 mice per group. SD: sleep deprivation.

performed at genus level. We performed analyses on all identified orders of bacteria in the samples. Among the 20 orders of bacteria clearly identified in our study, 6 orders had diurnal rhythmicity in control mice but this rhythmicity disappeared in the mice with sleep deprivation (Table 1). Fig 2 presented the diurnal rhythmicity changes in *o_bacteriodales* and *o_clostridiales*. These two types of bacteria were the most abundant ones that had diurnal rhythmicity in the control mice but not in the mice with sleep deprivation. These results suggest that sleep deprivation disrupts the diurnal rhythms of microbiota in the ileum.

**Sleep deprivation increased inflammatory cytokine in the blood and disrupted their diurnal rhythms**

Sleep deprivation increased the concentrations of IL6, IL-1β and IL-17 in the blood at certain time points (Fig 3). For example, sleep deprivation increased IL-6 at ZT21 compared with the corresponding results of control mice. In addition, sleep deprivation disrupted the diurnal rhythmicity of IL-6. The diurnal rhythms of IL-1β were maintained in the mice with sleep deprivation but sleep deprivation moved the peak concentrations of IL-1β from the daytime in the control mice to nighttime. There was no obvious rhythmicity in the blood IL-17 concentrations of both control mice and mice with sleep deprivation. In contrast, sleep deprivation decreased the concentrations of corticosterone in the blood and strengthened the diurnal rhythmicity of its concentrations (Fig 4).

Correlation analysis showed that multiple types of bacteria were positively associated with the concentrations of inflammatory cytokines. Interestingly, four genera of bacteria were positively associated with the concentrations of IL-17

**Table 1. Key parameters (mesor, amplitude, acrophase, and p value) of consinor analysis at order level of bacteria.**

| | Control | | | | SD | | | |
|---|---|---|---|---|---|---|---|---|
| | Mesor | Amplitude | Acrophase | P value | Mesor | Amplitude | Acrophase | P Value |
| Clostridiales | 47.0832 | 22.3784 | 14.1244 | 0.0187 | 53.6325 | 9.5804 | 10.3043 | 0.7108 |
| Lactobacillales | 34.5908 | 15.0297 | 0.0296 | 0.0630 | 26.0875 | 12.9276 | 22.2653 | 0.1487 |
| Bacteroidales | 14.5468 | 15.1304 | 4.5780 | 0.0019 | 13.7217 | 4.2704 | 10.3291 | 0.7581 |
| Deferribacterales | 2.6624 | 3.4205 | 15.7754 | 0.1468 | 1.4349 | 1.6006 | 14.2739 | 0.0088 |
| Pseudomonadales | 0.0356 | 0.0350 | 15.8463 | 0.0011 | 2.3579 | 1.5069 | 0.0154 | 0.6867 |
| Coriobacteriales | 0.4327 | 0.4889 | 2.2215 | 0.0001 | 0.5732 | 0.2475 | 10.7571 | 0.7756 |
| Enterobacteriales | 0.0445 | 0.0549 | 14.3317 | 0.0019 | 0.8537 | 0.7032 | 3.3541 | 0.2964 |
| Burkholderiales | 0.4876 | 0.2808 | 4.4209 | 0.3259 | 0.1925 | 0.1177 | 3.6256 | 0.3212 |
| Bifidobacteriales | 0.0002 | 0.0003 | 17.0000 | 0.2789 | 0.4948 | 0.1841 | 2.9731 | 0.8879 |
| Bacillales | 0.0424 | 0.0353 | 12.7161 | 0.1592 | 0.2190 | 0.2239 | 19.6041 | 0.0020 |
| Verrucomicrobiales | 0.0019 | 0.0027 | 17.3845 | 0.3911 | 0.2185 | 0.0725 | 0.0174 | 0.8812 |
| Flavobacteriales | 0.0291 | 0.0404 | 10.5355 | 0.1326 | 0.0990 | 0.1525 | 0.5547 | 0.3210 |
| Erysipelotrichales | 0.0207 | 0.0087 | 14.6313 | 0.7014 | 0.0517 | 0.0363 | 13.5356 | 0.3939 |
| Actinomycetales | 0.0030 | 0.0033 | 15.3599 | 0.0086 | 0.0187 | 0.0124 | 12.9565 | 0.4601 |
| Rhizobiales | 0.0008 | 0.0009 | 20.9587 | 0.2423 | 0.0206 | 0.0301 | 0.5503 | 0.3455 |
| Sphingobacteriales | 0.0005 | 0.0010 | 17.0000 | 0.2789 | 0.0140 | 0.0117 | 19.0139 | 0.4846 |
| Rhodospirillales | 0.0096 | 0.0113 | 11.1190 | 0.2939 | 0.0017 | 0.0035 | 1.0000 | 0.2789 |
| Sphingomonadales | 0.0054 | 0.0069 | 13.1534 | 0.3952 | 0.0028 | 0.0016 | 1.6680 | 0.7971 |
| Desulfovibrionales | **N/A** | **N/A** | **N/A** | **N/A** | 0.0032 | 0.0013 | 16.8404 | 0.8183 |
| Xanthomonadales | 0.0008 | 0.0016 | 13.0000 | 0.2789 | 0.0005 | 0.0009 | 17.0000 | 0.2789 |
| Other | 0.0018 | 0.0029 | 12.2865 | 0.0421 | 0.0017 | 0.0012 | 2.9187 | 0.3187 |

P values below 0.05 were marked in red.

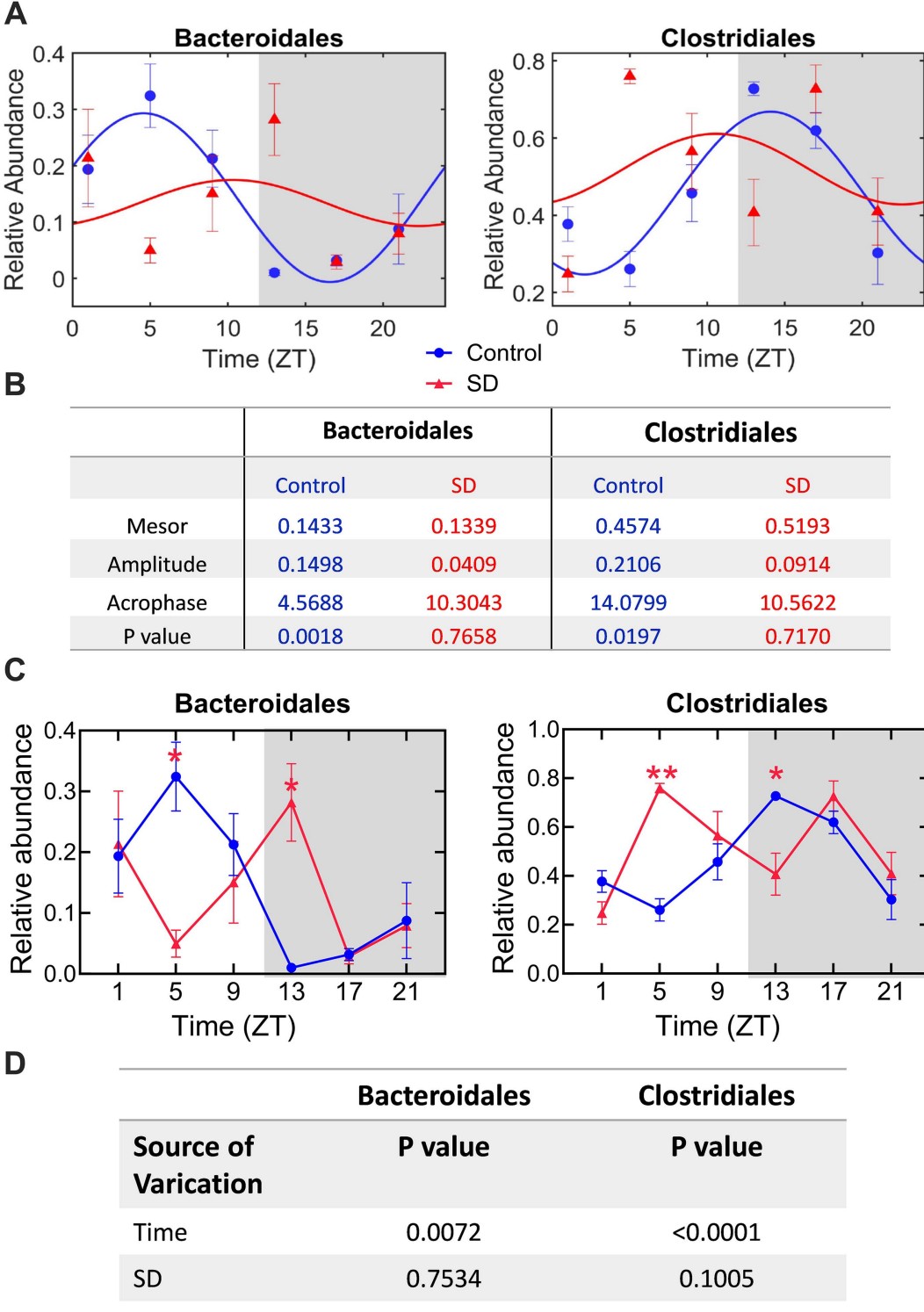

**Fig 2. Sleep deprivation disrupted the diurnal rhythmicity of gut microbiota at order level.** (A) graphical presentation of cosinor analysis. Light and dark phases of a day are shaded in white and gray, respectively. (B) results of cosinor analysis. (C) relative abundance of two types of bacteria. Light and dark phases of a day are shaded in white and gray, respectively. (D) results of two-way ANOVA. Data are presented in mean ± S.E.M. (n = 6). * P < 0.05, ** P < 0.001 compared with the corresponding control. SD: sleep deprivation.

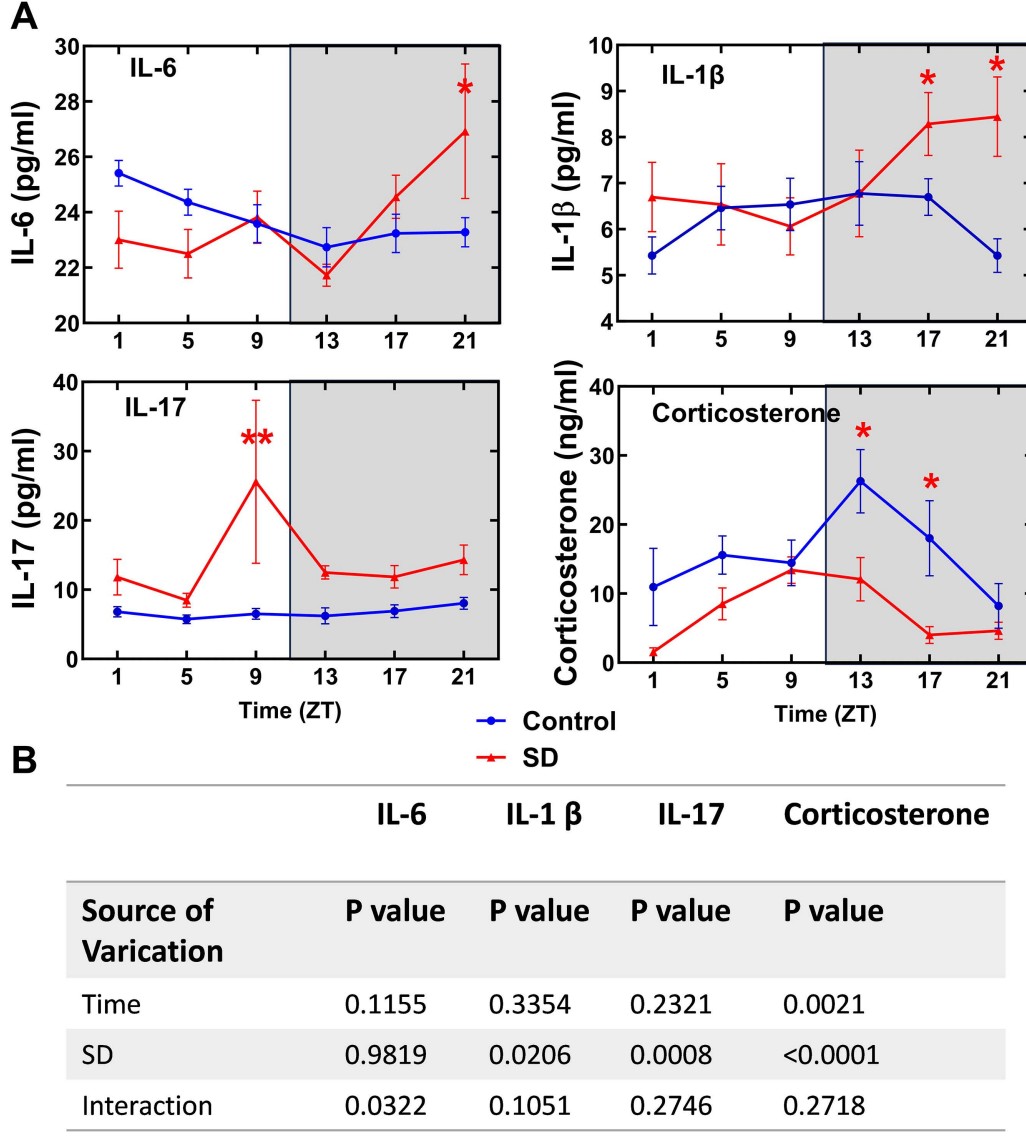

**B**

|  | IL-6 | IL-1 β | IL-17 | Corticosterone |
|---|---|---|---|---|
| **Source of Varication** | P value | P value | P value | P value |
| Time | 0.1155 | 0.3354 | 0.2321 | 0.0021 |
| SD | 0.9819 | 0.0206 | 0.0008 | <0.0001 |
| Interaction | 0.0322 | 0.1051 | 0.2746 | 0.2718 |

**Fig 3. Sleep deprivation increased inflammatory cytokines and decreased corticosterone in the blood.** (A) results of cytokines and corticosterone. Light and dark phases of a day are shaded in white and gray, respectively. (B) results of two-way ANOVA. Data are presented in mean ± S.E.M. (n = 6). * P < 0.05, ** P < 0.001 compared with the corresponding control (IL-6, IL-1β and IL-17 results) or sleep deprivation (corticosterone results) groups. SD: sleep deprivation.

but negatively associated with the concentrations of corticosterone. One of these four genera (*g_acinetobacter*) was also positively associated with IL-1β (Fig 5A).

## Sleep deprivation did not affect the concentrations of SCFAs in the blood

Sleep deprivation did not affect the concentrations of SCFAs in the blood (Fig 6). There was no obvious diurnal rhythmicity in the concentrations of SCFAs in control and sleep deprivation groups (Fig 7). Two genera of bacteria

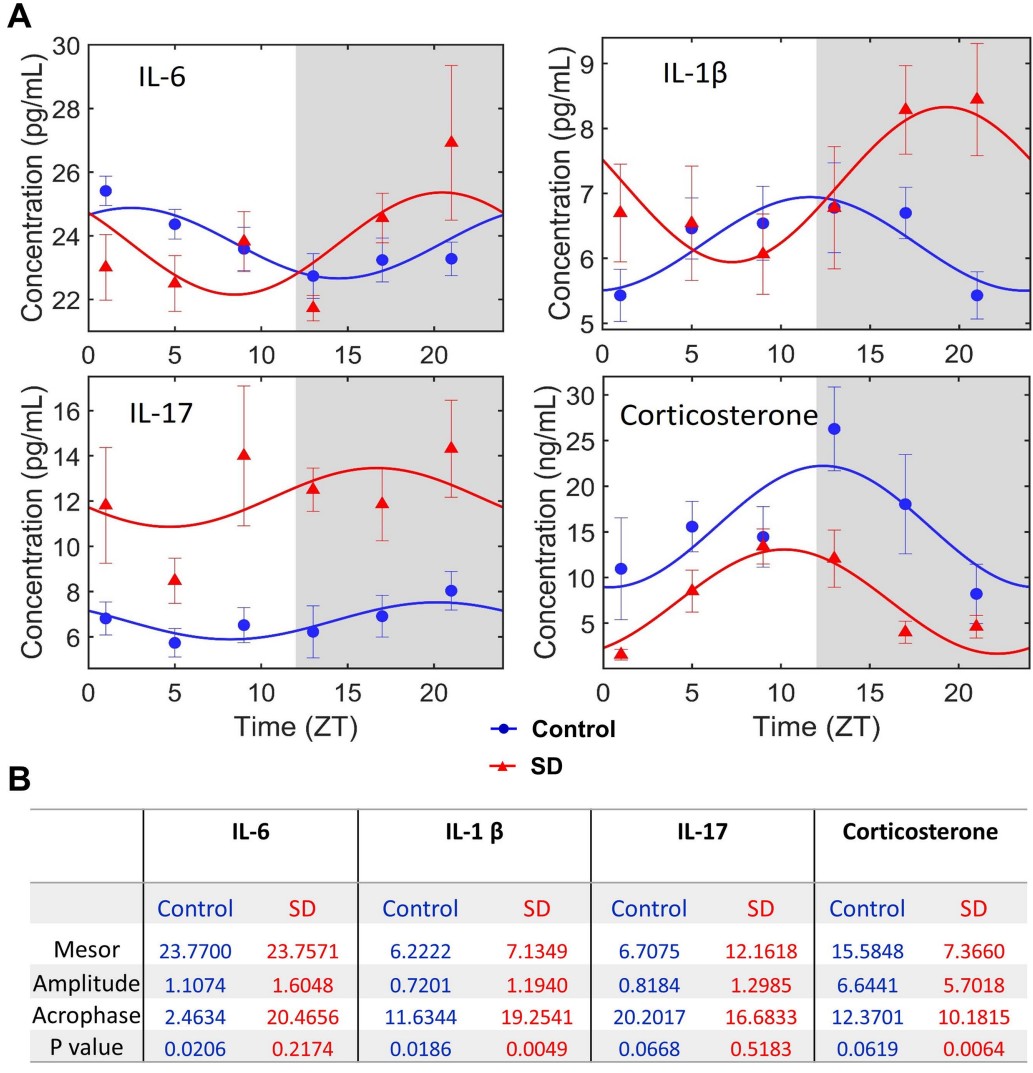

**Fig 4. Sleep deprivation disrupted the diurnal rhythmicity of inflammatory cytokines in the blood.** (A) graphical presentation of cosinor analysis. Light and dark phases of a day are shaded in white and gray, respectively. (B) results of cosinor analysis. Data are presented in mean±S.E.M. (n=6). SD: sleep deprivation.

(g_cupriavidus and g_exiguobacterium) were negatively associated with the concentrations of SCFAs (Fig 5B). One of them (g_exiguobacterium) was also negatively associated with the concentrations of corticosterone (Fig 5).

### Sleep deprivation increased the anxious behavior of mice

Mice with sleep deprivation spent more time in the corner area (297±7 s and 333±13 s, respectively, for control and sleep deprivation groups, P=0.0310) and less time in the border area (250±6 s and 221±9 s, respectively, for control and sleep deprivation groups, P=0.0109) than control mice during open field test (Fig 8), suggesting that mice with sleep deprivation are more anxious.

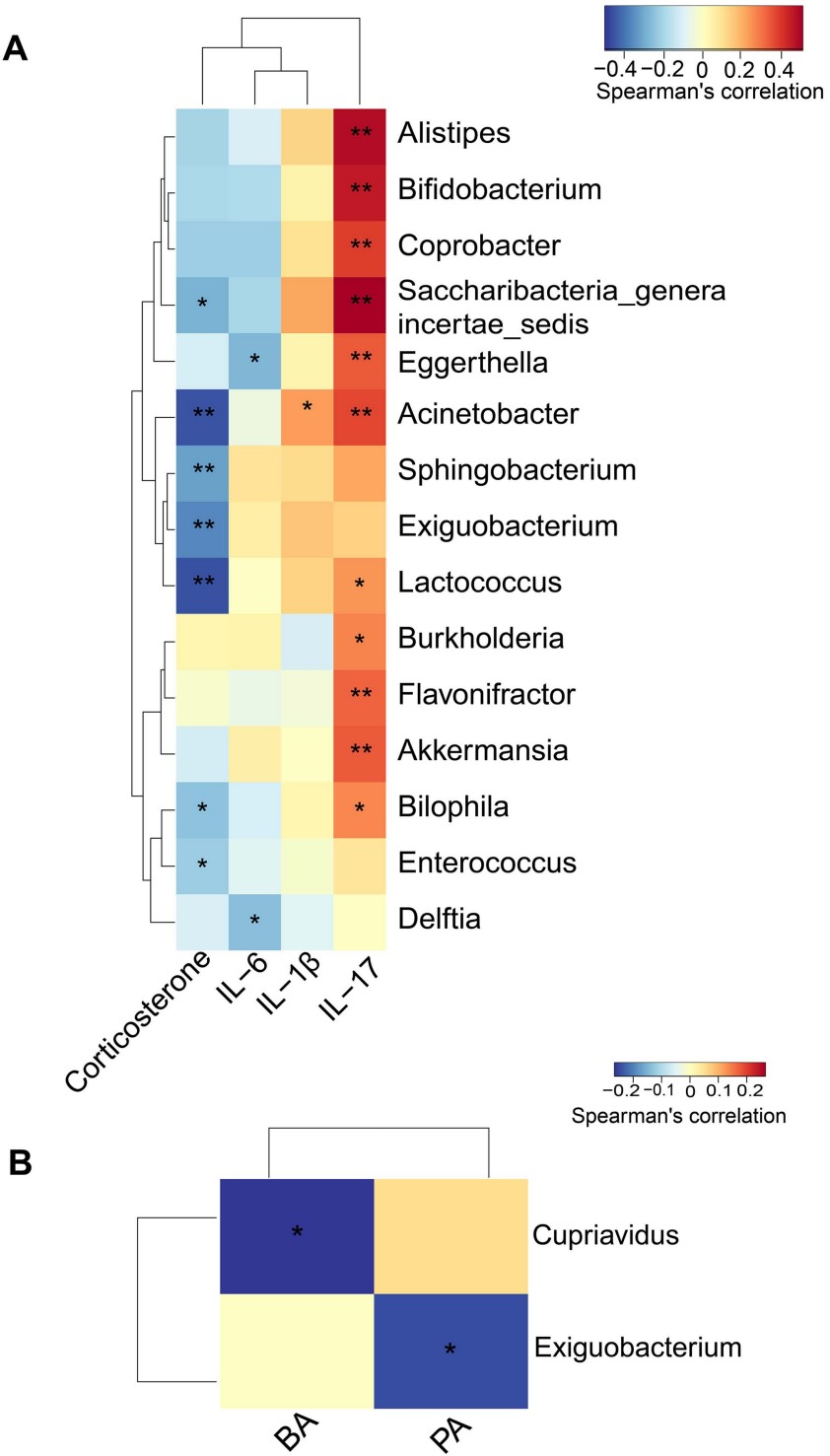

**Fig 5. Analysis of correlation between gut bacteria and blood cytokine concentrations or gut bacteria and short chain fatty acid concentrations.** (A) correlation between gut bacteria and blood cytokine concentrations. (B) correlation between gut bacteria and blood short chain fatty acid concentrations. PA: propionic acid, BA: butyric acid. * P < 0.05, ** P < 0.01 for the correlation.

## A

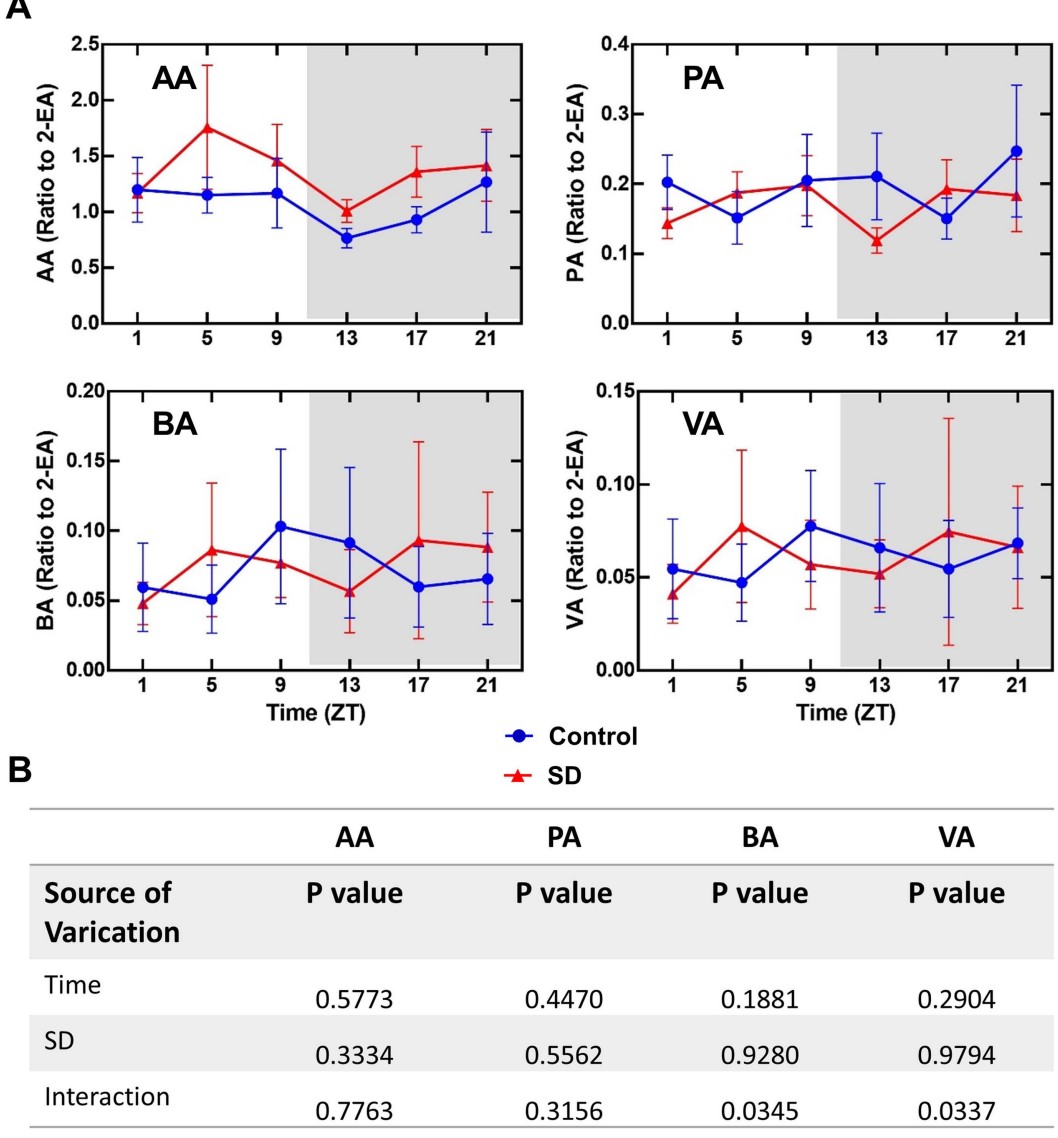

## B

|  | AA | PA | BA | VA |
|---|---|---|---|---|
| **Source of Varication** | **P value** | **P value** | **P value** | **P value** |
| Time | 0.5773 | 0.4470 | 0.1881 | 0.2904 |
| SD | 0.3334 | 0.5562 | 0.9280 | 0.9794 |
| Interaction | 0.7763 | 0.3156 | 0.0345 | 0.0337 |

**Fig 6. Sleep deprivation did not affect the concentrations of short chain fatty acids in the blood.** (A) results of short chain fatty acids. Light and dark phases of a day are shaded in white and gray, respectively. (B) results of two-way ANOVA. AA: acetic acid, PA: propionic acid, BA: butyric acid, VA: valeric acid. Data are presented in mean ± **S.**E.M. (n = 6). SD: sleep deprivation.

## Discussion

Our study has clearly shown the circadian rhythms of microbiota in the ileum of control mice. These results are consistent with previous studies [10,22]. However, most of the previous studies used feces as the samples to analyze gut microbiota [10–14]. It is easy for feces to be contaminated by skin and environmental microbiota during sampling. Bacteria in the feces may not directly contact the intestinal mucosa for the interaction between the host and the microbiota. In addition, feces are stored in the colon for various lengths of time and the bacteria in the feces are a mixture of microbiota from these times. Thus, it is a concern whether feces are good samples to be used for studying circadian rhythms. Also, since

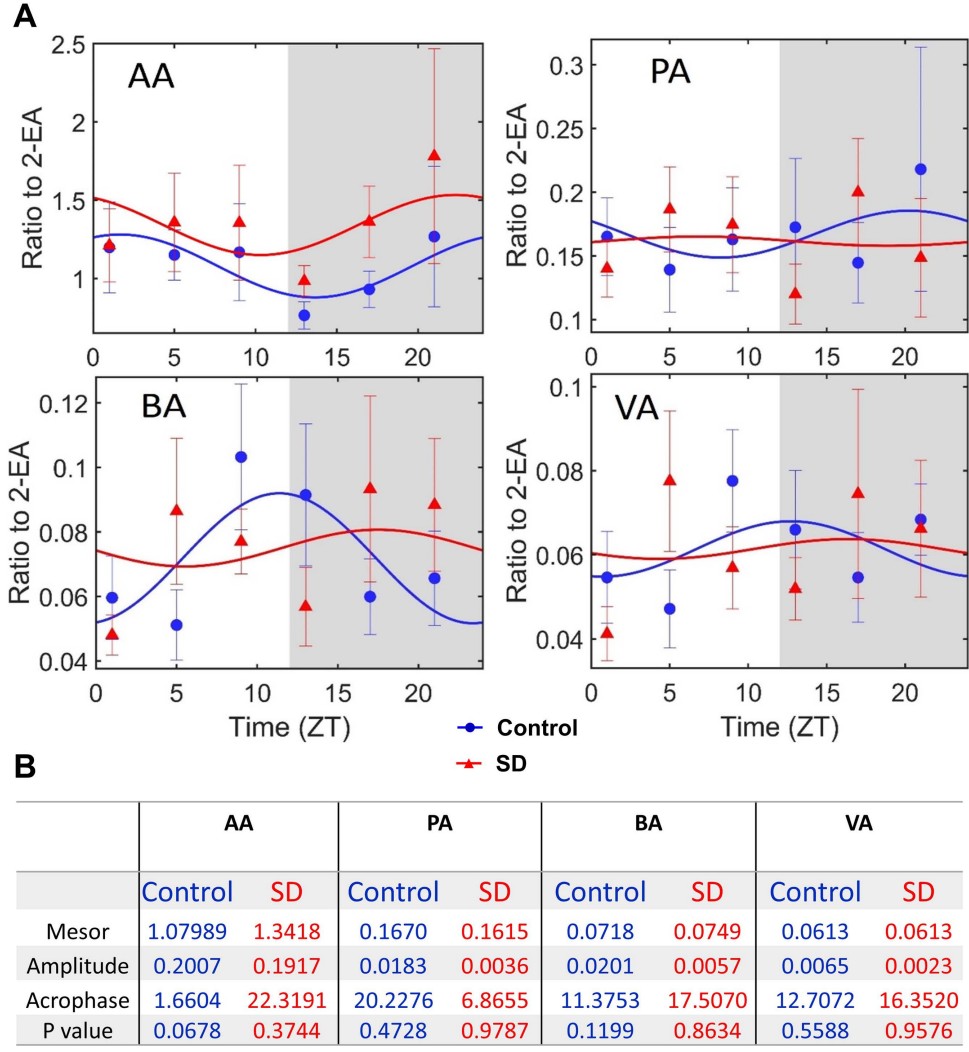

**Fig 7. No significant diurnal rhythmicity of short chain fatty acids in the blood was observed.** (A) graphical presentation of cosinor analysis. Light and dark phases of a day are shaded in white and gray, respectively. (B) results of cosinor analysis. AA: acetic acid, PA: propionic acid, BA: butyric acid, VA: valeric acid. Data are presented in mean±S.E.M. (n=6). SD: sleep deprivation.

| | AA | | PA | | BA | | VA | |
|---|---|---|---|---|---|---|---|---|
| | Control | SD | Control | SD | Control | SD | Control | SD |
| Mesor | 1.07989 | 1.3418 | 0.1670 | 0.1615 | 0.0718 | 0.0749 | 0.0613 | 0.0613 |
| Amplitude | 0.2007 | 0.1917 | 0.0183 | 0.0036 | 0.0201 | 0.0057 | 0.0065 | 0.0023 |
| Acrophase | 1.6604 | 22.3191 | 20.2276 | 6.8655 | 11.3753 | 17.5070 | 12.7072 | 16.3520 |
| P value | 0.0678 | 0.3744 | 0.4728 | 0.9787 | 0.1199 | 0.8634 | 0.5588 | 0.9576 |

gut microbiota is different at different locations in the intestine [23], our findings suggest that harvesting samples in the ileum is an excellent choice for studying diurnal rhythmicity of gut microbiota.

Our studies showed that sleep deprivation decreased the richness of microbiota and altered the taxonomic composition of the microbiota in the ileum. Previous studies have shown that sleep deprivation changes the relative richness of bacteria but does not alter the species makeup of gut microbiota as reflected by no changes in β diversity when compared with controls [24–26]. These previous studies used feces for the analysis of gut microbiota, which may be a reason why we detected a difference in β diversity between control mice and mice with sleep deprivation but previous studies did not show this difference. Furthermore, our study showed that sleep deprivation disrupted diurnal rhythmicity of gut microbiota. This finding presents a good example that disrupting the general circadian rhythms (sleep deprivation in our study) interrupts circadian rhythms of gut microbiota. Multiple mechanisms including altered feeding patterns and circadian clock gene expression in the brain and gut may contribute to our findings.

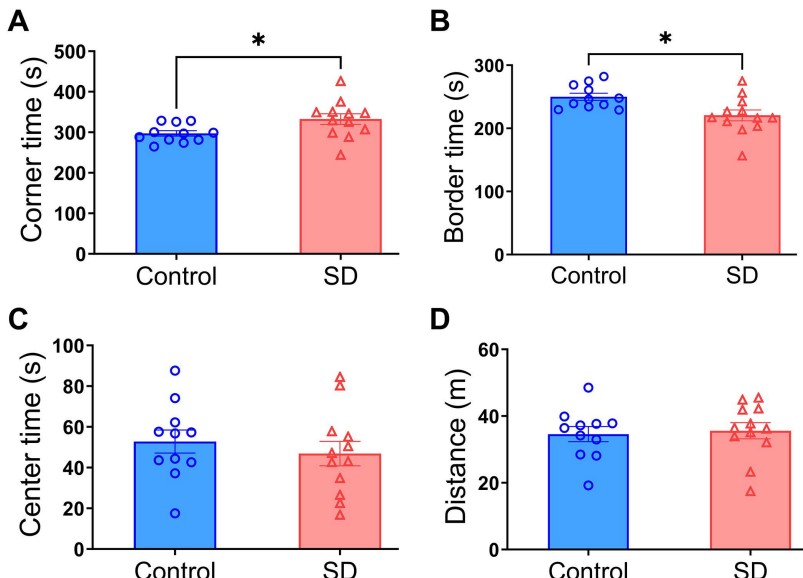

**Fig 8. Sleep deprivation increased the time staying in the corner area during open field test.** (A) time in the corner area. (B) time in the border area. (C) time in the center area. (D) total travel distance in the test. Data are presented in mean ± S.E.M. (n = 11 - 12) with the presence of individual animal result. * P < 0.05 for the comparisons. SD: sleep deprivation.

It is known that immune system has circadian changes. The changes include the concentrations of inflammatory cytokines and the number and functions of immune cells in the circulation [27–29]. Consistent with this knowledge, there was an obvious diurnal rhythmicity of IL-6 and IL-1β concentrations in the blood of control mice. However, sleep deprivation disrupted the circadian rhythms of IL-6. Importantly, there was an increase in the blood concentrations of IL-6, IL-17 and IL-1β in the mice with sleep deprivation. Gut microbiota can regulate immune responses [13,14]. Interestingly, multiple bacteria that were positively associated with the concentration of IL-17 in the blood were increased in the mice with sleep deprivation. These bacteria included *g_alistipes*, *g_bifidobacterium* and *g_acinetobacter* (Figs 1C and 5). These results suggest a cascade of sleep deprivation-gut microbiota changes-inflammatory cytokine increase. However, further studies are needed to determine whether the changes of blood cytokines in the mice with sleep deprivation are due to the disruption of diurnal rhythms of gut microbiota. Of note, mice with sleep deprivation had a decreased concentration of corticosterone, which may contribute to the increased concentrations of blood cytokines because corticosterone inhibits the expression of inflammatory cytokines [30].

SCFAs are mainly produced by gut microbiota and can mediate the effects of gut microbiota on immune responses, such as cytokine production [13]. In this study, there was no difference in the SCFA blood concentrations between control mice and mice with sleep deprivation. In addition, there were no diurnal rhythms in the blood concentrations of SCFAs of both control mice and mice with sleep deprivation except for a near significant P value (P = 0.0678) for acetic acid concentrations in the control mice. These results suggest that SCFAs may not be involved in the increased inflammatory cytokines and disruption of diurnal rhythmicity of these cytokines in the blood of mice with sleep deprivation. Our results of no obvious diurnal rhythmicity in the concentrations of SCFAs are different from a previous study that shows diurnal rhythmicity of SCFA concentrations in mice [31]. We measured SCFAs in the blood and the previous study determined SCFAs in the feces, which may contribute to the different findings.

In addition to characterizing the biochemical responses, the performance of mice in the open field test was assessed to determine the behavioral outcome of sleep deprivation. Mice with sleep deprivation were anxious, which was evidenced

by spending more time in the corner area than the control mice in the open field test [32]. The mechanisms for this finding are not clear. However, increased proinflammatory cytokines may contribute to the anxious status in mice with sleep deprivation because blood inflammatory cytokines, such as IL-6, is associated with psychiatric symptoms including depression and anxiety in man [33]. Inflammatory responses are also responsible for anxious behavior in mice with stress [34]. Thus, it is possible that increased inflammatory cytokines caused by sleep deprivation mediate the development of anxious behavior of mice in our study.

Our findings may have significant implications. Sleep deprivation induces gut microbiota changes in our study. Gut dysbiosis has been found in patients with various sleep disorders, sleep-wake cycle alteration and sleep architecture alteration, each of these conditions affecting millions of people in the United States [2,35,36]. Probiotic or synbiotic interventions improve the sleep quality and general health in adult participants with sub-health status [37]. Our results have shown a diurnal rhythmicity of gut microbiota. Although the role of this diurnal rhythmicity in regulating sleep is not clear, sleep is part of diurnal rhythmicity of humans and animals and there may be a close link between the gut microbiota rhythmicity and the overall body rhythmicity as supported by our findings that sleep deprivation disturbs the diurnal rhythmicity of gut microbiota. Thus, maintaining healthy gut microbiota with diurnal rhythmicity may be critical for the general health including having excellent sleep. Consistent with this possibility, fecal microbiota transplant with healthy microbiota improves autism-related symptoms in patients with autism [38] and sleep quality in patients with irritable bowel syndrome [39].

Our study has limitations. Our results showed diurnal rhythms in gut microbiota, which was disrupted by sleep deprivation. Also, sleep deprivation increased inflammatory cytokines and disrupted the circadian rhythms of cytokines. It is unclear whether the changed gut microbiota induces the alteration of cytokines and then the anxious status. Gut microbiota transplant study may be an approach to determine the relationship. However, the study may be difficult to perform because 1) harvesting microbiota in the ileum for transplant will need to sacrifice many animals to obtain enough samples, 2) it is hard to decide when to harvest the samples for transplant and when to transplant because of the diurnal rhythms of gut microbiota, and 3) it may not be possible to simulate the diurnal rhythms of gut microbiota and the disruption of these rhythms by using gut microbiota transplant.

In summary, our results showed diurnal rhythmicity in the ileum microbiota and blood inflammatory cytokines in mice. This rhythmicity was disrupted by sleep deprivation. Sleep deprivation also induced an anxious status in mice. These findings may have a significant and broad implication in human health considering the detrimental effects of circadian rhythm disruption and a large number of people suffering from sleep deprivation.

## Acknowledgments

The authors would like to thank Dr. Mingyan Guo of Zhiyi Zuo's group at the University of Virginia for helping collect samples after sleep deprivation. The whole study was performed at the University of Virginia.

## Author contributions

**Conceptualization:** Zhiyi Zuo.

**Data curation:** Weiran Shan.

**Formal analysis:** Weiran Shan, Wenzhe Zang, Zhiyi Zuo.

**Funding acquisition:** Zhiyi Zuo.

**Investigation:** Weiran Shan, Wenzhe Zang.

**Methodology:** Weiran Shan, Wenzhe Zang.

**Project administration:** Zhiyi Zuo.

**Resources:** Zhiyi Zuo.

**Software:** Wenzhe Zang.

**Supervision:** Zhiyi Zuo.

**Validation:** Weiran Shan, Wenzhe Zang.

**Visualization:** Weiran Shan, Wenzhe Zang, Zhiyi Zuo.

**Writing – original draft:** Weiran Shan, Wenzhe Zang, Zhiyi Zuo.

**Writing – review & editing:** Zhiyi Zuo.

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
