## [Decision Letter · Decision Letter 0]

22 Jul 2025

Dear Dr. Zhiyi Zuo,

Thank you for submitting your manuscript to PLOS ONE. After careful consideration, we feel that it has merit but does not fully meet PLOS ONE’s publication criteria as it currently stands. Therefore, we invite you to submit a revised version of the manuscript that addresses the points raised during the review process.

We look forward to receiving your revised manuscript.

Kind regards,

Huzaifa Umar

Academic Editor

PLOS ONE

Journal Requirements:

2. To comply with PLOS ONE submissions requirements, in your Methods section, please provide additional information regarding the experiments involving animals and ensure you have included details on (1) methods of anesthesia and/or analgesia, and (2) efforts to alleviate suffering.

5. In the online submission form, you indicated that [The data are available upon a reasonable request by contacting the corresponding author (Zhiyi Zuo at zz3c@virginia.edu).].

Reviewers' comments:

Reviewer's Responses to Questions

**Comments to the Author**

1. Is the manuscript technically sound, and do the data support the conclusions?

Reviewer #1: Yes

Reviewer #2: Yes

2. Has the statistical analysis been performed appropriately and rigorously?

Reviewer #1: Yes

Reviewer #2: Yes

3. Have the authors made all data underlying the findings in their manuscript fully available?

Reviewer #1: Yes

Reviewer #2: Yes

4. Is the manuscript presented in an intelligible fashion and written in standard English?

Reviewer #1: Yes

Reviewer #2: Yes

Reviewer #1: The study provides valuable insights into the gut microbiota–immune system interaction in sleep deprivation but requires improvements in data interpretation, clarity of methodology, and discussion depth.

1. Refine the title for conciseness while maintaining key elements; for example, Sleep Deprivation Disrupts Gut Microbiota Rhythms and Inflammatory Responses in Mice.

2. Improve grammar and specificity and include quantitative results in the abstract section.

3. Condense background information on circadian rhythms in the introduction section.

4. Explicitly state hypothesis and expected outcomes in the introduction section.

5. Strengthen rationale for behavioral assessments.

6. Provide details on microbiota sequencing normalization methods.

7. Clarify whether experimenters were blinded in behavioral assessments.

8. Ensure consistent use of statistical markers in figures and tables.

9. Include correlation analysis between cytokines and microbiota shifts.

10. Behavioral results should include effect size.

11. Improve figure clarity and readability.

12. Some speculative claims lack direct experimental validation (e.g., gut microbiota driving anxiety).

13. The clinical implications should be emphasized—how does this relate to human sleep disorders?

14. Lack of mechanistic validation (e.g., microbiota transplant studies).

15. Provide stronger mechanistic links between microbiota and behavior.

16. Expand discussion on clinical relevance.

Reviewer #2: he study employs a robust experimental design with comprehensive temporal sampling, enabling a detailed analysis of circadian disruptions caused by sleep deprivation. The integration of microbiome sequencing with cytokine profiling provides compelling evidence of altered microbial and immune rhythms, contributing valuable knowledge to the field of chronobiology and neuroimmunology. The findings have potential implications for understanding the health consequences of sleep loss and could inform therapeutic strategies aimed at mitigating circadian and inflammatory disturbances. Given the novelty, methodological rigor, and relevance to human health, I recommend acceptance of this manuscript for publication.

**Do you want your identity to be public for this peer review?** For information about this choice, including consent withdrawal, please see our Privacy Policy

Reviewer #1: No

Reviewer #2: No

---

## [Author Response · Author response to Decision Letter 1]

2 Sep 2025

Responses to the reviewers’ comments

We want to thank the reviewers for their effort in improving our paper. We have revised the manuscript based on their comments. The changes are marked in red in the manuscript for easy recognition. Our point-by-point responses to the reviewers’ comments are as follows.

Reviewer: 1

The study provides valuable insights into the gut microbiota–immune system interaction in sleep deprivation but requires improvements in data interpretation, clarity of methodology, and discussion depth.

Point 1: Refine the title for conciseness while maintaining key elements; for example, Sleep Deprivation Disrupts Gut Microbiota Rhythms and Inflammatory Responses in Mice.

Response: We have incorporated some suggested elements into the title of manuscript.

Point 2: Improve grammar and specificity and include quantitative results in the abstract section.

Response: We have gone through the abstract to improve its grammar and specificity. Also, we added interleukin 1β data.

Point 3: Condense background information on circadian rhythms in the introduction section.

Response: We have reduced the corresponding paragraph from 107 words to 96 words (paragraph 1 of page 3).

Point 4: Explicitly state hypothesis and expected outcomes in the introduction section.

Response: The hypotheses and expected outcomes are stated in the last paragraph of page 3.

Point 5: Strengthen rationale for behavioral assessments.

Response: The rationale for having behavioral assessments is now presented in the second paragraph of page 8.

Point 6: Provide details on microbiota sequencing normalization methods.

Response: The relevant information is now presented in the second paragraph of page 7.

Point 7: Clarify whether experimenters were blinded in behavioral assessments.

Response: Yes, the behavioral assessments were performed in a blind fashion. The information is in the second paragraph of page 8.

Point 8: Ensure consistent use of statistical markers in figures and tables.

Response: We have now used the same statistical markers in the figures.

Point 9: Include correlation analysis between cytokines and microbiota shifts.

Response: We performed correlation analysis between o_bacteroidales and cytokines or between o_clostridjales and cytokines. The P values for these correlations are all > 0.05 (see the table below this paragraph). We selected these two types of bacteria for correlation analysis because they had diurnal rhythmic changes. Our results do not suggest a correlation relationship between the bacteria and blood cytokines. Many reasons may explain this finding. For example, multiple types of bacteria may induce the production of one cytokine, but these bacteria may have different diurnal rhythmicity. Also, it may take various time lags between the changes of bacteria and cytokine production.

R2 P value

Bacteroidales

IL-1β

Control 0.02 0.77

SD 0.25 0.31

IL-6

Control 0.47 0.13

SD 0.31 0.25

IL-17

Control 0.21 0.36

SD 0.03 0.76

Clostridiales

IL-1β

Control 0.41 0.17

SD 0.004 0.91

IL-6

Control 0.37 0.20

SD 0.001 0.96

IL-17

Control 0.03 0.75

SD 0.01 0.87

Point 10: Behavioral results should include effect size.

Response: The effect size of sleep deprivation on mouse performance in the open field test is now presented in the second paragraph of page 13.

Point 11: Improve figure clarity and readability.

Response: We have enlarged the labels that were small in the previous version of figures and saved them in tiff file with high resolution.

Point 12: Some speculative claims lack direct experimental validation (e.g., gut microbiota driving anxiety).

Response: There was one place that had this speculative claim. We have modified the corresponding sentences in the Conclusions of Abstract not to have such a claim.

Point 13: The clinical implications should be emphasized—how does this relate to human sleep disorders?

Response: We have added a paragraph to discuss the relevance of our findings to human sleep disorders in the second paragraph of page 16.

Point 14: Lack of mechanistic validation (e.g., microbiota transplant studies).

Response: We agree that there is a lack of mechanistic validation for the link between disruption of diurnal rhythmicity and anxious behavior of mice. Microbiota transplant experiments may seem to be an approach for the validation. However, these experiments are not practical. First, we measured the diurnal rhythmicity of gut microbiota in the ileum. Harvesting enough materials to prepare gut microbiota from ileum for transplant and making sure that the transplant is successful is very difficult because the donors and recipients have to be euthanized for harvesting the samples. Simulating the diurnal rhythmicity of gut microbiota and the disruption of the rhythmicity is not possible to achieve. We have expanded the discussion in the third paragraph of page 16.

Point 15: Provide stronger mechanistic links between microbiota and behavior.

Response: Please see our response to your point 14. We have added discussion in the first paragraph and the last paragraph of page 15 and the first paragraph of page 16 to provide evidence for the possible link between gut microbiota and anxious behavior.

Point 16: Expand discussion on clinical relevance.

Response: A paragraph is added to discuss the clinical relevance of our findings (second paragraph of page 16).

Reviewer: 2

The study employs a robust experimental design with comprehensive temporal sampling, enabling a detailed analysis of circadian disruptions caused by sleep deprivation. The integration of microbiome sequencing with cytokine profiling provides compelling evidence of altered microbial and immune rhythms, contributing valuable knowledge to the field of chronobiology and neuroimmunology. The findings have potential implications for understanding the health consequences of sleep loss and could inform therapeutic strategies aimed at mitigating circadian and inflammatory disturbances. Given the novelty, methodological rigor, and relevance to human health, I recommend acceptance of this manuscript for publication.

Response: Thank you very much for your recommendation and encouragement.

---

## [Editor Report · Decision Letter 1]

16 Oct 2025

Sleep deprivation disrupts diurnal rhythmicity of gut microbiota and blood inflammatory cytokines in mice

PONE-D-25-03292R1

Dear Dr. Zhiyi,

We’re pleased to inform you that your manuscript has been judged scientifically suitable for publication and will be formally accepted for publication once it meets all outstanding technical requirements.

Within one week, you’ll receive an email detailing the required amendments. When these have been addressed, you’ll receive a formal acceptance letter, and your manuscript will be scheduled for publication.

Kind regards,

Huzaifa Umar

Academic Editor

PLOS ONE
---

## [Editor Report · Acceptance letter]

PONE-D-25-03292R1

PLOS ONE

Dear Dr. Zuo,

I'm pleased to inform you that your manuscript has been deemed suitable for publication in PLOS ONE. Congratulations! Your manuscript is now being handed over to our production team.

Kind regards,

on behalf of

Dr. Huzaifa Umar

Academic Editor

PLOS ONE